# The association of telomere length and telomerase activity with adverse outcomes in older patients with non-ST-elevation acute coronary syndrome

**Danny Chan**[1,2], **Carmen Martin-Ruiz**[3], **Gabriele Saretzki**[4], **Dermot Neely**[5], **Weiliang Qiu**[6], **Vijay Kunadian**[1,2]*

**1** Translational and Clinical Research Institute, Faculty of Medical Sciences, Newcastle University, Newcastle upon Tyne, United Kingdom, **2** Cardiothoracic Centre, Freeman Hospital, Newcastle upon Tyne Hospitals NHS Foundation Trust, Newcastle upon Tyne, United Kingdom, **3** BioScreening Facility, Newcastle University, Newcastle upon Tyne, United Kingdom, **4** Ageing Biology Centre and Institute for Cell and Molecular Biosciences, Newcastle University, Newcastle upon Tyne, United Kingdom, **5** Department of Biochemistry, Newcastle upon Tyne Hospitals NHS Foundations Trust, United Kingdom, **6** Sanofi Genzyme, Framingham, MA, United States of America

* vijay.kunadian@newcastle.ac.uk

**Data Availability Statement:** There are ethical restrictions on sharing a de-identified data set, as data contain potentially sensitive information as

## Abstract

### Background

Non-ST elevation acute coronary syndrome (NSTEACS) occurs more frequently in older patients with an increased occurrence of recurrent cardiac events following the index presentation. Telomeres are structures consisting of repeated DNA sequences as associated shelterin proteins at the ends of chromosomes. We aim to determine whether telomere length (TL) and telomerase activity (TA) predicted poor outcomes in older patients presenting with NSTEACS undergoing invasive care.

### Method

Older patients undergoing invasive management for NSTEACS were recruited to the ICON-1 biomarker study (NCT01933581). Peripheral blood mononuclear cells (PBMC) were recovered on 153 patients. DNA was isolated and mean TL was measured by quantitative PCR expressed as relative T (telomere repeat copy number) to S (single copy gene number) ratio (T/S ratio), and a telomere repeat amplification assay was used to assess TA during index presentation with NSTEACS. Primary clinical outcomes consisted of death, myocardial infarction (MI), unplanned revascularisation, stroke and significant bleeding recorded at 1 year. TL and TA were divided into tertile groups for analysis. Cox proportional hazards regression was performed. Ordinal regression was performed to evaluate the relationship between TL and TA and traditional cardiovascular risk factors at baseline.

### Results

298 patients were recruited in the ICON-1 study of which 153 had PBMC recovered. The mean age was 81.0 ± 4.0 years (64% male). Mean telomere length T/S ratio was 0.47 ±

imposed by study sponsor and consent for data sharing not obtained from patients. Data are available upon request to the study Sponsor tnu-tr. sponsormanagement@nhs.net.

**Funding:** The research is supported by the National Institute for Health Research (NIHR) Newcastle Biomedical Research Centre based at Newcastle-upon-Tyne Hospitals NHS Foundation Trust and Newcastle University. VK has received research funding from the British Heart Foundation (CS/15/7/31679). WQ is salaried by Sanofi Genzyme. The funder provided support in the form of salaries for authors VK and WQ but did not have any additional role in the study design, data collection and analysis, decision to publish, or preparation of the manuscript. The specific roles of these authors are articulated in the 'author contributions' section.

**Competing interests:** The authors have declared that no competing interests exist. VK has received research funding from NIHR BRC Newcastle, AstraZeneca, and British Heart Foundation. WQ is salaried by Sanofi Genzyme. This does not alter our adherence to PLOS ONE policies on sharing data and materials.

0.25 and mean TA was 1.52 ± 0.61 units. The primary composite outcome occurred in 44 (28.8%) patients. There was no association between short TL or low TA and incidence of the primary composite outcome (Hazard Ratio [HR] 1.50, 95% Confidence Interval [CI] 0.68–3.34, $p = 0.32$ and HR 1.33, 95% CI 0.52–3.36, $p = 0.51$ respectively).

## Conclusion

TL and TA are not found to be associated with the incidence of adverse outcomes in older patients presenting with NSTEACS undergoing invasive care.

## Clinical trial registration

URL: https://www.clinicaltrials.gov Unique identifier: NCT01933581

## Introduction

Older age is a well-known cardiovascular disease (CVD) risk factor, especially for coronary artery disease (CAD)[1–3]. In a rapidly progressing ageing population, CAD prevalence, and the related detrimental consequences, can only be expected to increase. Non ST-elevation acute coronary syndromes (NSTEACS) are more common within the older population, with the UK's Myocardial Ischaemia National Audit Project (MINAP) data showing that 46% of all non ST elevation myocardial infarction (NSTEMIs) suffered between 2006 and 2010 occurred in patients aged ≥75 years old[4].

Telomeres are structures of tandemly repeated hexanucleotide TTAGGG sequences associated with specific shelterin proteins at the end of eukaryotic chromosomes. They protect internal chromosomal regions of DNA from degradation during cell division and gradually shorten with each cycle due to the end replication problem as well as the sensitivity to oxidative stress[5]. At a certain point the telomeres become too short to facilitate cell division, resulting in cell senescence or apoptosis. Telomere length (TL) and telomerase activity (TA) have been investigated regarding their possible applicability as biomarkers for age-related chronic diseases, including CVD. Shorter TL has also been linked to an increased risk of adverse events in patients with pre-existing CAD[6]. These studies have mostly been restricted to younger patients, resulting in a paucity of research investigating this relationship in older patients. Therefore we sought to investigate the association of TL and TA with adverse outcomes in older patients presenting with NSTEACS undergoing invasive management.

## Methods

### Study design

The Improve Cardiovascular Outcomes in high-risk older patients with acute coronary syndrome (ICON1) study is a multicentre prospective cohort study which aimed to develop a risk score for high-risk older adults, the FRAIL-HEART score.[7–9] The study protocol has been published previously[10]. The current study is a planned study as outlined in the the ICON1 study protocol[10]. Older patients (aged ≥ 65 years old) presenting with NSTEACS with planned invasive management were recruited from two tertiary cardiac care hospitals. The study was carried out in accordance with the Declaration of Helsinki, and approved by the regional ethics committee (NRES Committee North East–Sunderland 12/NE/0160). Participants gave written informed consent to take part in the study. The *a priori* primary outcome

was created from data on death, acute coronary syndrome (ACS), unplanned revascularisation, stroke and BARC (Bleeding Academic Research Consortium)-defined bleeding (type 2 or greater) at 1 year[11]. Follow-up was carried out in research clinics, and for patients unable to attend telephone consultations were carried out. In participants where more than one component of the composite outcome occurred, time-to-first-event was used. Detailed information on baseline assessment had been previously published.[10] Frailty was assessed using Fried Frailty Index, derived from the Cardiovascular Health Study with a score of 0 classed as robust, 1–2 as pre-frail and ≥3 as frail.[12]

## Telomere and telomerase analysis

Peripheral blood mononuclear cells were isolated using leucosep tubes, and DNA extraction performed using QIAamp® DNA mini kit (Qiagen). DNA concentration and quality was ascertained using 260/280 nm spectrophotometry (NanoDropTM). Samples were diluted to 10 ng/ul prior to TL assay. Analysis of mean TL was performed using a quantitative, real-time polymerase chain reaction-based assay, in quadruplicate using a 7900 HT Fast Real Time PCR system (Applied Biosciences™). Comparison between signals from T (telomere repeat copy number) to S (single copy gene number) were made, with the relative T/S ratio calculated. The intra-assay coefficient of variation was 2.7%, and the inter-assay coefficient of variation was 5.1%. Telomerase activity was assessed using a telomere repeat amplification assay (Telo-TAGGG™; Roche, Switzerland)[13] using 500ng lysate from PBMCs isolated in the same way as above.

## Statistical analysis

Statistical analyses were conducted with the use of SPSS software version 22 (SPSS Inc., Chicago, IL, USA) and a two-tailed-test p value <0.05 was considered statistically significant. Those variables classed as normally distributed were reported as mean ± standard deviation (SD). Variables that were non-normally distributed were reported as median [interquartile range (IQR)]. For TL analysis, data were split into three equal tertiles with patients categorized into long TL (LTL, ratio ≥0.5), medium TL (MTL, ratio 0.3468–0.5), and short TL (STL, ratio ≤ 0.3467). TA data were also split into three equal tertiles for analysis: high TA (≥ 1.86 units), mid TA (1.32–1.86 units), and low TA (≤ 1.31 units).

Cox proportional hazard models were used to model the risk of incidence of the primary composite end-point by tertile of TL and TA. Differences in baseline characteristics were assessed between tertiles with one-way ANOVA for normally distributed continuous data, Kruskal-Wallis testing for non-normally distributed continuous data, and chi-squared test ($\chi2$) or Fisher exact test (as appropriate with expected cell counts of <5) for categorical variables. Baseline characteristics did not vary between tertiles of TL or TA (see **Table 1**) therefore univariate cox proportional hazard models were used. In addition, adjusted Cox proportional hazards regression models were also used, adjusting for age, gender and frailty. Kaplan-Meier survival analysis was performed to compare event-free survival from incidence of the primary composite outcome between tertiles of TL and TA, probed with the Log-rank test. TL and TA was also assessed as continuous variables (S1–S3 Figs). The mean TL and TA subjects with composite events were compared to subjects without composite events with Wilcoxon rank sum test. In addition TL and TA was also classified into two groups based on receiver operating characteristic curve to determine the best cut-off. Details can be found in **S1 Table**.

**Table 1. Baseline characteristics.**

| | Total (N = 135) | LTL (N = 43) | MTL (N = 47) | STL (N = 45) | p-value |
|---|---|---|---|---|---|
| **Demographics** | | | | | |
| Age, years (SD) | 80.8 (4.1) | 80.2 (3.4) | 80.7 (4.4) | 81.6 (3.9) | 0.39 |
| Female, *n* (%) | 47 (34.8) | 20 (46.5) | 12 (25.5) | 15 (33.3) | 0.49 |
| NSTEMI, *n* (%) | 246 (82.0) | 31 (72.1) | 39 (83.0) | 36 (80.0) | 0.43 |
| UA, *n* (%) | 54 (18.0) | 12 (27.9) | 8 (17.0) | 9 (20.0) | 0.44 |
| **Clinical Measures** | | | | | |
| BMI, kg m$^{-2}$ (SD) | 26.7 (4.2) | 26.7 (4.7) | 26.7 (3.9) | 26.7 (4.5) | 0.95 |
| Systolic BP, mmHg (SD) | 145.2 (26.5) | 147.1 (19.9) | 145.5 (26.5) | 142.8 (32.3) | 0.80 |
| NYHA functional class, *n* (%): | | | | | |
| 1—no limitation of activity | 51 (38.1) | 16 (38.1) | 21 (44.7) | 14 (31.1) | 0.18 |
| 2—slight limitation of activity | 50 (37.3) | 18 (42.9) | 15 (31.9) | 17 (37.8) | |
| 3—marked limitation of activity | 32 (23.9) | 3 (9.1) | 21(31.3) | 8 (23.5) | |
| 4—unable to carry out activity | 1 (0.7) | 1 (2.4) | 0 (0.0) | 0 (0.0) | |
| GRACE Score, points (SD) | 130.2 (18.5) | 130.2 (17.1) | 129.8 (18.5) | 130.7 (20.2) | 0.97 |
| **Medical History** | | | | | |
| Diabetes, *n* (%) | 30 (22.2) | 13 (30.2) | 7 (14.9) | 10 (22.2) | 0.64 |
| Hypertension, *n* (%) | 92 (68.1) | 25 (58.1) | 35 (74.5) | 32 (71.1) | 0.30 |
| Hyperlipidaemia, *n* (%) | 69 (51.1) | 23 (53.5) | 25 (53.2) | 21 (46.7) | 0.53 |
| Renal impairment, *n* (%) | 33 (24.4) | 9 (20.9) | 15 (31.9) | 9 (20.0) | 0.39 |
| Previous MI, *n* (%) | 37 (27.4) | 12 (27.9) | 11 (23.4) | 14 (31.1) | 0.47 |
| Previous angina, *n* (%) | 57 (42.2) | 18 (41.9) | 21 (44.7) | 18 (40.0) | 0.85 |
| Previous PCI, *n* (%) | 24 (17.8) | 7 (16.3) | 11 (23.4) | 6 (13.3) | 0.86 |
| Previous CABG, *n* (%) | 8 (5.9) | 4 (9.3) | 1 (2.1) | 3 (6.7) | 0.07 |
| HF, *n* (%) | 10 (7.4) | 3 (7.0) | 4 (8.5) | 3 (6.7) | 0.82 |
| AF, *n* (%) | 19 (14.1) | 4 (9.3) | 8 (17.0) | 7 (15.6) | 0.36 |
| PVD, *n* (%) | 13 (9.6) | 4 (9.3) | 4 (8.5) | 5 (11.1) | 0.86 |
| Previous TIA/Stroke, *n* (%) | 19 (14.1) | 3 (7.0) | 9 (19.1) | 7 (15.6) | 0.63 |
| Osteoarthritis, *n* (%) | 4 (3.0) | 1 (2.3) | 1 (2.1) | 2 (4.4) | 0.46 |
| Peptic ulcer disease, *n* (%) | 6 (4.4) | 2 (4.7) | 1 (2.1) | 3 (6.7) | 0.85 |
| COPD, *n* (%) | 23 (17.0) | 6 (14.0) | 11 (23.4) | 6 (13.3) | 0.51 |
| Malignancy, *n* (%) | 11 (8.1) | 3 (7.0) | 6 (12.8) | 2 (4.4) | 0.43 |
| Bleeding problems, *n* (%) | 4 (3.0) | 0 (0.0) | 2 (4.3) | 2 (4.4) | 0.47 |
| Anaemia, *n* (%) | 11 (8.1) | 2 (4.7) | 5 (10.6) | 4 (8.9) | 0.45 |
| **Smoking Status** | | | | | |
| Current smoker, *n* (%) | 9 (6.8) | 1 (2.4) | 4 (8.5) | 4 (8.9) | 0.36 |
| Ex-smoker, *n* (%) | 66 (49.6) | 20 (48.8) | 25 (53.2) | 21 (46.7) | 0.93 |
| Never-smoker, *n* (%) | 58 (43.6) | 20 (48.8) | 18 (38.3) | 20 (44.4) | 0.94 |
| **Frailty Indices** | | | | | |
| **Fried index components** | | | | | |
| Shrinking criterion, *n* (%) | 24 (17.8) | 4 (9.5) | 10 (21.3) | 10 (22.2) | 0.36 |
| Low physical endurance, *n* (%) | 33 (24.6) | 6 (14.3) | 15 (31.9) | 12 (26.7) | 0.14 |
| Low physical activity, *n* (%) | 35 (25.9) | 6 (14.3) | 15 (31.9) | 14 (31.1) | 0.34 |
| Weakness, *n* (%) | 83 (62.4) | 25 (59.5) | 28 (60.9) | 30 (66.7) | 0.80 |
| Slow walking speed, *n* (%) | 23 (17.0) | 7 (16.7) | 7 (15.6) | 9 (20.0) | 0.85 |
| **Rockwood score**, *n* (%) | | | | | |
| 1 – 2 | 46 (34.1) | 18 (41.9) | 13 (27.7) | 15 (33.3) | 0.28 |

*(Continued)*

**Table 1.** (Continued)

| | Total (N = 135) | LTL (N = 43) | MTL (N = 47) | STL (N = 45) | p-value |
|---|---|---|---|---|---|
| 3 – 4 | 72 (53.3) | 22 (51.2) | 27 (57.4) | 23 (51.1) | |
| 5 – 7 | 17 (12.6) | 3 (7.0) | 7 (14.9) | 7 (15.6) | |
| **Quality of Life and Co-morbidity** | | | | | |
| MoCA, points (SD) | 25.1 (3.4) | 25.5 (2.5) | 25.4 (3.6) | 24.2 (3.8) | 0.29 |
| SF-36 PCS, points (SD) | 36.6 (11.3) | 35.4 (8.9) | 37.6 (12.2) | 35.8 (11.7) | 0.98 |
| SF-36 MCS, points (SD) | 49.6 (10.2) | 47.6 (10.7) | 50.0 (10.2) | 50.9 (9.8) | 0.91 |
| Health state, % (SD) | 63.7 (19.4) | 59.8 (19.8) | 66.2 (19.7) | 62.3 (18.0) | 0.88 |
| Charlson index, points (SD) | 5.3 (1.7) | 5.4 (1.5) | 5.1 (1.8) | 5.6 (1.9) | 0.63 |
| **Blood results** | | | | | |
| Haemoglobin, g L$^{-1}$ (SD) | 12.9 (1.9) | 13.0 (1.6) | 13.2 (1.7) | 12.4 (2.5) | 0.88 |
| Creatinine, μmol L$^{-1}$ (SD) | 103.5 (32.7) | 106.7 (39.0) | 99.5 (28.4) | 108.3 (34.1) | 0.60 |
| Total cholesterol, mmol L$^{-1}$ (SD) | 4.3 (1.3) | 4.5 (1.5) | 4.3 (1.3) | 3.9 (0.8) | 0.71 |
| hsCRP, mg L$^{-1}$ (SD) | 12.3 (37.6) | 7.5 (6.7) | 15.8 (50.9) | 9.5 (16.1) | 0.29 |
| Troponin T, ng L$^{-1}$ (SD) | 511.8 (983.7) | 606.0 (1159.1) | 403.6 (860.9) | 636.8 (1035.8) | 0.55 |
| eGFR, mL min$^{-1}$ 1.73 m$^{-2}$ (SD) | 54.5 (19.5) | 54.3 (21.2) | 56.2 (18.1) | 51.2 (20.5) | 0.71 |

Abbreviations: AF—atrial fibrillation, BMI—body mass index, BP—blood pressure, CABG—coronary artery bypass graft, COPD—chronic obstructive pulmonary disease, EQ—EuroQol form, GRACE—Global Registry of Acute Coronary Events, HF—heart failure, IDAOPI—Income Deprivation Affecting Older People Index, IMD—Index of Multiple Deprivation, IQR—interquartile range, LTL- long telomere length, MTL- medium telomere length, STL-short telomere length, MCS—mental component summary, MI—myocardial infarction, MoCA—Montreal Cognitive Assessment, NYHA—New York Heart Association class, PCI—percutaneous coronary intervention, PCS—physical component summary, PVD—peripheral vascular disease, SD—standard deviation, SF-36—short form 36, TIA—transient ischaemic attack, eGFR—estimated glomerular filtration rate, hsCRP—C-reactive protein

## Results

### Baseline characteristics

A total of 153 patients had PBMC samples recovered as part of the ICON1 biomarker sub-study and of these samples, 135 patients had appropriate samples for TL analysis and 67 samples were analysed for TA. Patient baseline characteristics are illustrated in **Table 1**. There was no significant difference in baseline characteristics between tertiles of TL.

The mean age of participants was 81.0 ± 4.0 years and 64.0% were male, with 87.5% of participants classed as pre-frail or frail. The mean cohort telomere length was 0.47 ± 0.25 and was split into tertiles for analysis: LTL (n = 43, mean 0.74 ± 0.27), MTL (n = 47, mean 0.42 ± 0.05) and STL (n = 45, mean 0.25 ± 0.06). The mean cohort TA was 1.52 ± 0.61 units and split into tertiles for analysis: high TA (n = 23, 2.09 ± 0.24), mid TA (n = 22, 1.61 ± 0.15) and low TA (n = 22, 0.83 ± 0.48).

Of the 153 patients in the study, 111 patients (82.2%) underwent PCI. As demonstrated in **Table 2**, there were no significant differences in the coronary arteries affected, number of stents used or medical therapy on discharge between tertiles of TL.

### Telomere length, telomerase activity and adverse outcomes

Incidence of the primary composite outcome at 1 year is detailed in **Table 3**. The primary composite outcome occurred in 44 patients (28.8%). There was no significant difference in the incidence of the primary composite outcome between tertiles of TL (p = 0.57) or TA (p = 0.62).

**Table 2. Procedural characterisitics.**

| | Total (N = 135) | LTL (N = 43) | MTL (N = 47) | STL (N = 45) | p-value |
|---|---|---|---|---|---|
| **Killip on admission** | | | | | |
| 1, *n* (%) | 115 (81.2) | 35 (81.4) | 41 (87.2) | 39 (86.7) | 0.79 |
| 2, *n* (%) | 16 (11.8) | 7 (16.3) | 4 (8.5) | 5 (11.1) | |
| 3, *n* (%) | 4 (0.03) | 1 (2.3) | 2 (4.3) | 1 (2.2) | |
| **PCI, *n* (%)** | 111 (82.2) | 34 (79.1) | 40 (85.1) | 37 (82.2) | 0.76 |
| **Coronary arteries affected** | | | | | |
| One vessel disease, *n* (%) | 85 (63.0) | 26 (60.5) | 29 (61.7) | 30 (66.7) | 0.81 |
| Multi-vessel disease, *n* (%) | 27 (20.0) | 9 (20.9) | 11 (23.4) | 7 (15.6) | 0.63 |
| LMS, *n* (%) | 4 (3.0) | 0 (0.0) | 2 (4.3) | 2 (4.4) | 0.38 |
| LAD, *n* (%) | 64 (47.4) | 19 (44.2) | 23 (48.9) | 22 (48.9) | 0.88 |
| LCx, *n* (%) | 34 (25.2) | 9 (20.9) | 12 (25.5) | 13 (28.9) | 0.69 |
| RCA, *n* (%) | 39 (28.9) | 14 (32.6) | 15 (31.9) | 10 (22.2) | 0.48 |
| **Number of stents used** | | | | | |
| 0, *n* (%) | 27 (20.0) | 11 (25.6) | 7 (14.9) | 9 (20.0) | 0.78 |
| 1, *n* (%) | 55 (40.7) | 14 (32.6) | 22 (46.8) | 19 (42.2) | |
| 2, *n* (%) | 30 (22.2) | 11 (25.6) | 10 (21.3) | 9 (20.0) | |
| 3, *n* (%) | 13 (9.6) | 6 (14.0) | 4 (8.5) | 3 (6.7) | |
| 4, *n* (%) | 7 (5.2) | 1 (2.3) | 3 (6.4) | 3 (6.7) | |
| 5, *n* (%) | 2 (1.5) | 0 (0.0) | 1 (2.1) | 1 (2.2) | |
| 6, *n* (%) | 1 (0.7) | 0 (0.0) | 0 (0.0) | 1 (2.2) | |
| **Medical therapy on discharge** | | | | | |
| Aspirin, *n* (%) | 132 (97.8) | 41 (95.3) | 46 (97.9) | 45 (100.0) | 0.33 |
| Clopidogrel, *n* (%) | 78 (57.8) | 25 (58.1) | 26 (55.3) | 27 (60.0) | 0.90 |
| Prasugrel, *n* (%) | 0 (0.0) | 0 (0.0) | 0 (0.0) | 0 | |
| Ticagrelor, *n* (%) | 52 (38.5) | 16 (37.2) | 19 (40.4) | 17 (37.8) | 0.94 |
| Statin, *n* (%) | 126 (93.3) | 42 (97.7) | 42 (89.4) | 42 (93.3) | 0.98 |
| ACEi/A2RB, *n* (%) | 117 (86.7) | 36 (83.7) | 42 (89.4) | 39 (86.7) | 0.73 |
| BB, *n* (%) | 113 (83.7) | 37 (86.0) | 42 (89.4) | 34 (75.6) | 0.18 |
| CCB, *n* (%) | 42 (31.1) | 12 (27.9) | 15 (31.9) | 15 (33.3) | 0.85 |
| ISMN, *n* (%) | 42 (31.1) | 15 (34.9) | 16 (34.0) | 11 (24.4) | 0.49 |
| Nicorandil, *n* (%) | 20 (14.8) | 8 (18.6) | 7 (14.9) | 5 (11.1) | 0.61 |
| PPI, *n* (%) | 63 (46.7) | 20 (46.5) | 16 (34.0) | 27 (60.0) | 0.055 |
| Warfarin, *n* (%) | 10 (7.4) | 2 (4.7) | 4 (8.5) | 4 (8.9) | 0.70 |
| DOAC, *n* (%) | 5 (3.7) | 2 (4.7) | 2 (4.3) | 1 (2.2) | 0.80 |

Abbreviations: ACEi/A2RB–angiotensin converter inhibitor/angiotensin 2 receptor blocker, BB–beta blocker, CCB–calcium channel blocker, DOAC—direct acting oral anticoagulants, ISMN–isosorbide mononitrate, LAD–left anterior desceding coronary artery disease, LCx–left coronary artery disease, LMS–left main stem coronary artery disease, LTL- long telomere length, MTL- medium telomere length, STL-short telomere length, PPI–proton pump inhibitor, RCA–right coronary artery disease

Cox proportional hazard models between tertiles of TL and TA and the incidence of the primary composite outcome are detailed in **Table 4**. There was an increased hazard of the primary composite outcome in patients with a MTL (hazard ratio HR 1.28, 95% Confidence Interval CI 0.56–2.91, $p = 0.56$) and STL (HR 1.50, 95% CI 0.68–3.34, $p = 0.32$) when compared to the reference patients with a LTL, however this was not statistically significant. There was no significant difference between TA and the incidence of the primary composite outcome in patients with a mid TA (HR 0.90, 95% CI 0.94–2.60, $p = 0.89$) and low TA (HR 1.33, 95% CI 0.52–3.36, $p = 0.51$). The results are consistent after adjusting for age, gender and frailty in the

**Table 3. Outcome measures.**

| 1-year Outcomes | Total (N = 135) | LTL (N = 43) | MTL (N = 47) | STL (N = 45) | p-value |
|---|---|---|---|---|---|
| Composite, n (%) | 38 (28.1) | 10 (23.3) | 13 (27.7) | 15 (33.3) | 0.57 |
| Death, n (%) | 6 (4.4) | 1 (2.3) | 2 (4.2) | 3 (6.7) | 0.61 |
| Myocardial infarction, n (%) | 12 (8.9) | 5 (11.6) | 4 (8.4) | 3 (6.7) | 0.71 |
| Unplanned revascularisation, n (%) | 6 (4.4) | 4 (9.3) | 0 | 2 (4.4) | 0.10 |
| Stroke, n (%) | 2 (1.5) | 1 (2.3) | 0 | 1 (2.2) | 0.58 |
| Significant bleeding, n (%) | 21 (15.6) | 4 (9.3) | 9 (19.1) | 8 (17.8) | 0.54 |
| All-cause re-hospitalisation, n (%) | 69 (51.1) | 21 (48.8) | 25 (53.2) | 23 (51.1) | 0.92 |

LTL long telomere length, MTL medium telomere length, STL short telomere length.

analysis model. Kaplan-Meier survival curves for TL (Fig 1) and TA (Fig 2) show no difference between groups for both TL (p = 0.61) and TA (p = 0.74). These results are consistent when primary outcome were analysed without BARC. See S2 Table.

## Discussion

This is the first study to examine the association of TL and TA with adverse clinical outcomes in older patients with NSTEACS undergoing an invasive treatment strategy. Neither TL nor TA were found to be associated with adverse outcomes in older patients with NSTEACS.

Ischaemic heart disease (IHD) is the leading cause of death worldwide[14] and importantly, the demographic of the condition is changing, with increasing frequency of older patients presenting with IHD. Studies have also shown that advancing age is associated with increased risk of mortality following cardiovascular events[15]. The increase of IHD burden in older patients,

**Table 4. Telomere length and telomerase activity as predictors of the primary composite outcome.**

| | Hazard ratio | 95% confidence interval | p-value |
|---|---|---|---|
| **Model 1** | | | |
| TL* | | | |
| MTL | 1.28 | 0.56–2.91 | 0.56 |
| STL | 1.50 | 0.68–3.34 | 0.32 |
| TA† | | | |
| Mid | 0.90 | 0.94–2.60 | 0.89 |
| Low | 1.33 | 0.52–3.36 | 0.51 |
| Model 2 | | | |
| TL* | | | |
| MTL | 1.49 | 0.64–3.45 | 0.35 |
| STL | 1.58 | 0.71–3.55 | 0.26 |
| TA† | | | |
| Mid | 1.04 | 0.37–2.92 | 0.94 |
| Low | 1.18 | 0.46–3.06 | 0.73 |

Univariate (model 1) and multivariate (model 2; adjusted for age, gender and frailty) Cox regression analysis preformed for combined and primary outcomes alone using telomere length and telomerase activity as predictors. Both predictors were divided into tertiles for analysis.

* LTL used as reference.

† High used as reference. MTL- medium telomere length, STL-short telomere length and TA-telomerase activity.

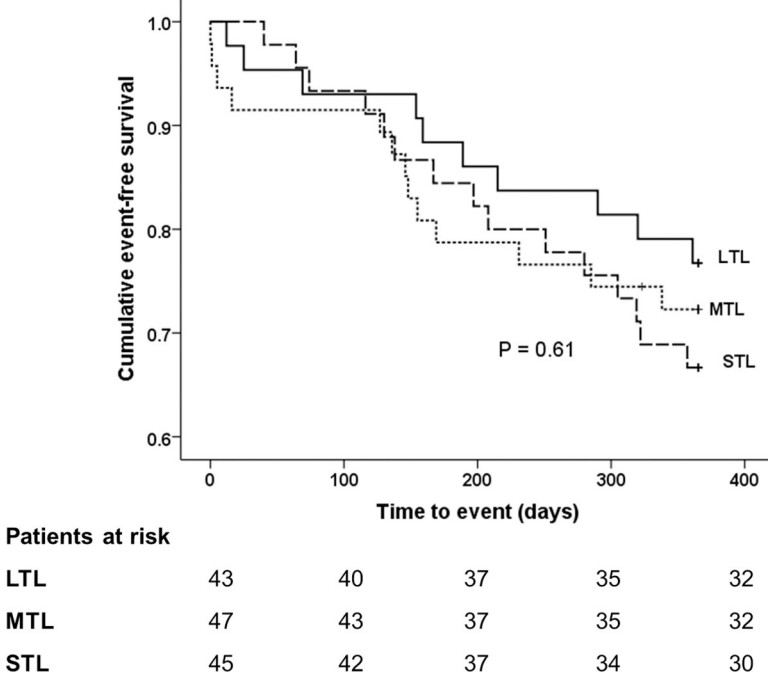

**Fig 1. Telomere length and cumulative event rates.** Cumulative event-free survival from the composite primary end-point by tertile of telomere length. LTL ≥ 0.5 T/S ratio, MTL 0.3468 to 0.5 T/S ratio and STL ≤ 0.3467 T/S ratio. P value from the Log-rank test. LTL, long telomere length; MTL; medium telomere length and STL; short telomere length.

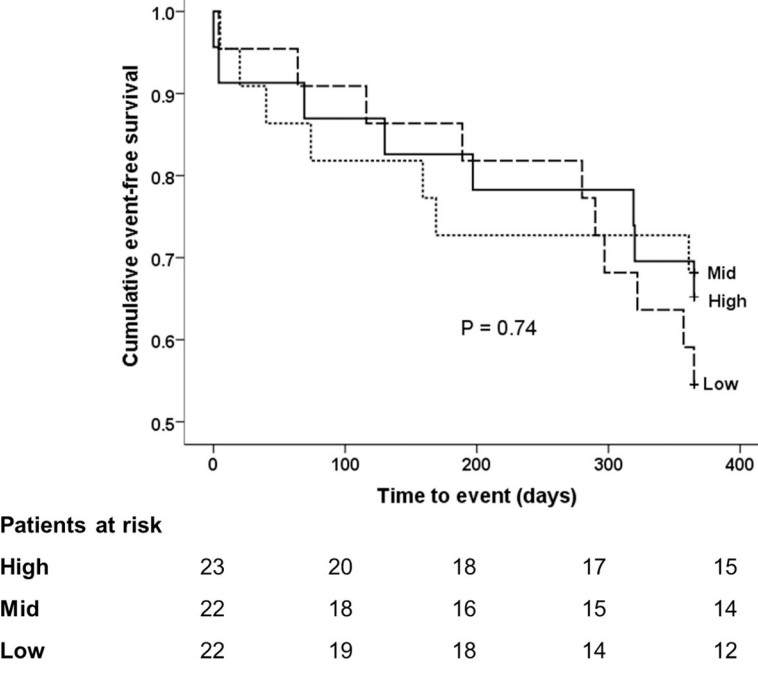

**Fig 2. Telomerase activity and cumulative event rates.** Cumulative event-free survival from the composite primary end-point by tertile of telomerase activity. High telomerase activity denotes ≥1.86 units, mid telomerase activity 1.32 to 1.86 units, and low telomerase activity ≤1.31 units. P value from the Log-rank test.

together with the increased risk of poorer outcome, drives a need for markers which could guide the management of ACS in this group of patients.

Telomeres are protective structures at the end of chromosomes which act to stabilise genome integrity, and have been implicated as a marker of biological age and age-related diseases. The length of telomere shortens with subsequent cell division, and this shortening is accelerated by inflammation and oxidative stress, processes implicated in the pathophysiology of IHD. To prevent telomere shortening to a critical length, in cells such as germ cells and stem cells, telomerase activity is high to maintain and elongate telomeres. In most somatic cells, telomerase activity is absent or low, but a mouse study has suggested that telomerase might play a role in regulating tissue repair in cardiomyocytes and endothelial progenitor cells [16].

Previous studies evaluating the association of telomere lenth and telomerase activity on clinical outcomes in different populations have shown varied results[17–25]. In a prospective WOSCOPS study of 484 participants with a mean age of 55 years, participants in the middle and the lowest tertiles of telomere length were more at risk of developing a coronary heart disease event than were individuals in the highest tertile (odds ratio [OR] for coronary heart disease: 1.51, 95% CI 1.15–1.98; p = 0.0029 in the middle tertile; 1.44, 1.10–1.90, p = 0.0090 in the lowest)[26]. In a meta-analysis of prospective and retrospective studies with 5566 participants with CHD, the pooled relative risk for CHD of shortest versus longest tertile telomere length was 1.54 (95% CI 1.30 to 1.83)[27]. Moreover, in a previous study of 170 patients with stable angina or acute coronary syndrome and using virtual histology intravascular ultrasound, it was shown that shorter TL was associated with a high risk of unstable plaques (calcified thin-capped fibroatheroma), with an OR of 1.24 (95% CI 1.01 to 1.53)[28]. Our results contradict the previous literature, a finding which may be partly explained by differences in the study design, methodology and patient cohort. Our study population is generally older, underwent invasive management, and we specifically examined NSTEACS as this is the predominant ACS phenotype in this age group. This differs to previous research, which was in a majority performed in a younger cohort and included all ACS types.

Epel and colleagues[25] found that the rate of telomere shortening was predictive of mortality from cardiovascular disease in elderly men, suggesting that low telomerase activity contribute to cardiovascular risk. However, our study found that telomerase activity was not associated with adverse outcomes following invasive management of NSTEACS. This may partly be explained by the difference of study populations. Epel et al. recruited healthy older volunteers prior to cardiovascular event, compared to the patients presenting with NSTEACS in the current study. Hence, this study focused on telomere length in those whom already developed coronary artery disease. The results together would suggest that dependent upon the cumulative oxidative stress and inflammatory burden, telomere attrition may play a different role in the process of atherosclerosis in older patients. Consistent with this hypothesis, Perez-Rivera et al[29]. found worse prognosis with short telomere length in middle aged men with ACS but not in older patients.

It has been speculated that telomere attrition and the resulting cell senescence may act as a mechanism for restricting atheromas[30]. Indeed, this is consistent with our study result showing low telomerase activity was not associated with adverse outcomes in older patients. Furthermore, telomerase-deficient mice TERC$^{-/-}$ ApoE$^{-/-}$ had fewer atherosclerotic lesions compared to TERC$^{+/+}$ ApoE$^{-/-}$[31], and Willeit et al[32] showed that TL was a risk predictor for myocardial infarction, but not for *de novo* stable angina and intermittent claudication. Therefore, telomere attrition causes senescence and apoptosis, but may have process-specific effect on atherosclerosis[30].

Recently Werner and colleagues found that specific exercise modalities were associated with increased telomere length and telomerase activity[33].This raises the interesting possibility that the benefits of exercise in reducing and preventing frailty in the older population may act through regulating cellular senescence. The current study found no association between telomere length and frailty, which is consistent with the results demonstrated by Brault et al. in older adults with cardiac disease[34]. These results suggest that frailty may be the consequence of repeated physiological stress induced premature cellular senescence in addition to progressive telomere shortening. It would be interesting to investigate the effect of exercise on telomere length and frailty in older adults with established cardiac disease.

## Limitations

The limitation of this study is the relatively small number of adverse outcomes observed in the 1 year follow-up, therefore these results should be interpreted with caution. Recruiting older patients to clinical research is very challenging[35, 36]. Very few studies have evaluated older patients with NSTEACS. A lot of factors and therapy that show positive correlation in younger patients, don't quite show this in older patients. Given the growth of older patients with coronary disease, it is important the cardiovascular community are aware of such differences in the best care of older patients. Our study with the sample size included with detailed statistical analysis provides important information regarding these patients.

Larger follow-up studies would be required to evaluate the true value of TL and TA as predictors in this setting. A further limitation is that we have not considered leukocyte subpopulations distributions, a factor that can determine the measurements of both TL and TA TL[37] as well as the association of both parametres to cardiac dysfunction[38].

## Conclusion

The present study found TL and TA were not associated with adverse outcomes in older patients presenting with NSTEACS.

## Supporting information

**S1 Fig. Scatter plot of telomere length vs. age.** Scatter plot analysis of TL with age yields a non statstiacally significant increasing regression line in aged 75–80 (Spearman's correlation coefficient, $r_s$ = 0.052, p = 0.66) and a decreasing regression line in aged > 80 ($r_s$ = -0.11, p = 0.34).
(TIF)

**S2 Fig. Scatter plot of telomerase activity vs. age.** Scatter plot analysis of TA with age yields a non-statistically significant decreasing regression line (Spearman's correlation coefficient, $r_s$ = -0.22, p = 0.071).
(TIF)

**S3 Fig. Boxplots of telomere length for subjects with primary outcomes vs without primary outcomes.** A Wilcoxon signed–rank test showed that telomere lengths did not elicit a significant change for subjects with primary outcomes compared to subjects without composite events (Z = -0.655, p = 0.071).
(TIF)

**S4 Fig. Boxplots of telomerase activity for subjects with primary outcomes vs without primary outcomes.** A Wilcoxon signed–rank test showed that telomerase activity did not elicit a significant change for subjects with primary outcomes compared to subjects without

composite events (Z = -1.274, p = 0.20).
(TIF)

**S1 Table. Telomere length and telomerase activity classified by ROC, as predictors of the primary composite outcome.** Cox regression analysis preformed for combined and primary outcomes alone using telomere length and telomerase activity as predictors. Both predictors were divided into two groups for analysis based upon area under curve (ROC). For TL, area under curve was 0.57 (p = 0.17). The best cutoff was 0.61 with sensitivity of 87% and specificity of 80%. With this cut-off, 111 participants had STL (82.2%) and 24 had LTL (17.8%). The area under curve was also measured for TA at 0.54 (p = 0.57), with cut-off of 1.88 (sensitivity of 80% and specificity of 64%). 47 (70.1%) participants classified as low TA and 20 (29.9) as high TA. * LTL used as reference. † High used as reference. TL- telomere length and TA-telomerase activity.
(DOCX)

**S2 Table. Telomere length and telomerase activity as predictors of the primary composite outcome without major bleeding.** Cox regression analysis preformed for combined outcomes without major bleeding using telomere length and telomerase activity as predictors. Both predictors were divided into tertiles for analysis. * LTL used as reference. † High used as reference. MTL- medium telomere length, STL-short telomere length and TA-telomerase activity.
(DOCX)

## Acknowledgments

The authors would like to thank:

Dr. J Ahmed, Dr. A Bagnall, Dr. R Das, Dr. R Edwards, Dr. M Egred, Dr. I Purcell, Professor. I Spyridopoulos and Professor. A Zaman of Freeman Hospital, Newcastle upon Tyne for their help with data collection.

Cardiology CRN research team at Freeman Hospital: Mrs. Kathryn Proctor and Mrs. Jennifer Adams-Hall for their support with follow-up of study patients.

Dr. Mark de Belder and Mrs. Bev Atkinson, the James Cook University Hospital, South Tees Hospitals NHS Foundation Trust, Middlesbrough, United Kingdom for their help with data collection.

The authors would like to thank Clinical research fellows, Dr H. Sinclair, Dr M. Veerasamy, Dr. J Batty, Dr. B Beska of Freeman Hospital, Newcastle upon Tyne for their hard work in patient recruitment and data collection

## Author Contributions

**Conceptualization:** Vijay Kunadian.

**Data curation:** Carmen Martin-Ruiz, Gabriele Saretzki, Vijay Kunadian.

**Formal analysis:** Carmen Martin-Ruiz, Gabriele Saretzki, Weiliang Qiu, Vijay Kunadian.

**Funding acquisition:** Vijay Kunadian.

**Investigation:** Carmen Martin-Ruiz, Vijay Kunadian.

**Methodology:** Vijay Kunadian.

**Project administration:** Vijay Kunadian.

**Resources:** Vijay Kunadian.

**Software:** Vijay Kunadian.

**Supervision:** Vijay Kunadian.

**Validation:** Vijay Kunadian.

**Writing – original draft:** Danny Chan, Carmen Martin-Ruiz, Gabriele Saretzki, Dermot Neely, Weiliang Qiu, Vijay Kunadian.

**Writing – review & editing:** Danny Chan, Carmen Martin-Ruiz, Gabriele Saretzki, Dermot Neely, Weiliang Qiu, Vijay Kunadian.

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
