## [Decision Letter · Decision Letter 0]

16 Oct 2019

PONE-D-19-23361

The association of telomere length and telomerase activity with adverse outcomes in older patients with non-ST-elevation acute coronary syndrome

PLOS ONE

Dear Dr Kunadian,

Thank you for submitting your manuscript to PLOS ONE. After careful consideration, we feel that it has merit but does not fully meet PLOS ONE’s publication criteria as it currently stands. Therefore, we invite you to submit a revised version of the manuscript that addresses the points raised during the review process.

We would appreciate receiving your revised manuscript by Nov 30 2019 11:59PM. To enhance the reproducibility of your results, we recommend that if applicable you deposit your laboratory protocols in protocols.io, where a protocol can be assigned its own identifier (DOI) such that it can be cited independently in the future. For instructions see: http://journals.plos.org/plosone/s/submission-guidelines#loc-laboratory-protocols

We look forward to receiving your revised manuscript.

Kind regards,

Gianluigi Savarese

Academic Editor

PLOS ONE

Journal Requirements:

The research is supported by the National Institute for Health Research (NIHR) Newcastle Biomedical Research Centre based at Newcastle-upon-Tyne Hospitals NHS Foundation Trust and Newcastle University. VK has received research funding from the British Heart Foundation (CS/15/7/31679). The views expressed are those of the authors and not necessarily those of the NHS, the NIHR or the Department of Health.

Please remove any funding-related text from the manuscript and let us know how you would like to update your Funding Statement. Currently, your Funding Statement reads as follows: Nil

4.  We note that one or more of the authors are employed by a commercial company: Sanofi Genzyme.

6. We note you have included a table to which you do not refer in the text of your manuscript. Please ensure that you refer to Tables 1, 2 and 3 in your text; if accepted, production will need this reference to link the reader to the Table.

Additional Editor Comments:

The Authors performed unadjusted cox regression models due to the lack of statistically significant difference between tertiles of TL. However, the sample size is small and this could led not to identify any statistically significant difference in baseline characteristics. Thus, I suggest a stepwise approach to be able to provide adjusted models, although incl. a limited number of potential confounders.

Reviewers' comments:

Reviewer's Responses to Questions

**Comments to the Author**

1. Is the manuscript technically sound, and do the data support the conclusions?

Reviewer #1: Yes

Reviewer #2: Yes

2. Has the statistical analysis been performed appropriately and rigorously? 

Reviewer #1: Yes

Reviewer #2: Yes

3. Have the authors made all data underlying the findings in their manuscript fully available?

Reviewer #1: Yes

Reviewer #2: Yes

4. Is the manuscript presented in an intelligible fashion and written in standard English?

Reviewer #1: Yes

Reviewer #2: Yes

5. Review Comments to the Author

Reviewer #1: 1. Introduction: Previous data is well summarized and the main aim of the study is clearly specified.

2. Methods:

-Patients were selected from ICON1 study. Authors affirmed that patients aged > 65 years old were included. Maybe this cut-off point does not reflect the real “old patients”, in addition mean age of the patients included in this study is around 80 years old. Did you selected very old patients from the whole sample of ICON1 study or you included any patients older than 65 years old?

-You divided the sample in tertiles. This is completely correct but it might be more precise to do it according to the best cut off point calculated by ROC curve. In this way you can provide data of sensitivity and specificity.

-You designed a composite endpoint with thrombotic and bleeding events. In previous studies, telomere length seems to be connected to atherosclerosis and the evidence about its relationship with bleeding is weaker. It could be interesting to calculate a secondary combined endpoint formed only by thrombotic events.

3. Results

-The incidence of adverse events is too low in your patients. Previous studies of very old patients with ACS showed 6 months mortality around 12% (Alegre et al. J Am Med Dir Assoc. 2017 Nov 17).

-Please, comment the percentage of AMI and unstable angina in your sample.

-Could you provide the incidence of mortality caused by cardiovascular causes in your sample?

-Do you have data about the characteristics of the coronary event of your patients (number of affected arteries, Killip class, number of stents used…)? Is the peak value the troponin measurement provided en table 1?

4. Discussion

--In the reference 25 of your manuscript (Epel et al.), telomere attrition but not initial telomere length was associated to cardiovascular mortality. In your study you did not find any relationship between telomerase activity and mortality. Could you explain in detail how your results agree or disagree with this paper?

-Other papers have studied the prognosis effect of telomere length in older patients with ACS (Am J Cardiol 2014;113:418e421). In this paper telomere length showed a prognostic effect in middle aged men with ACS but not in older patients. The group of elderly patients in this study was small (n: 52), presented STEMI and NSTEMI and revascularization was performed only in the 53% of patients so the sample is quite different to yours. Nevertheless, results are similar so it might be interesting to discuss this paper in your manuscript.

Reviewer #2: The Authors presented a substudy of the ICON-1 Biomarker study, aiming to determine the possible Association between telomere length and telomerase activity with the outcome of old patients presenting with NSTEACS.

The manuscript is well written and addresses a clinically relevant issue with potential translational relevance.

However, there are major issues to be resolved before considering the paper subitable for pubblication.

- was this substudy pre-specified in the trial design? If not, this could be a relevant methodological bias.

- are the patients recruited consecutive?

- the reason why almost half of the patients recruited in the general study have been esclude in this substudy is clear and not reported. Please add. It is also relevant to explain Why Only in 67 patients telomerase activity data were available. These may be considered profound bias in the analysis of the data presented.

- The Authors stated that these patients underwent an invasive management of the NSTEACS. How many of these underwent PCI? Procedural data are not reported. Therapy on admission and at discharge was not reported. All these informations might have a profound impact on the data and outcome analysis.

- the division in to tertiles seems to be arbitrary.

- When considering frail vs pre-frail patients, are there any difference in the distribuition of TL and TA and their association with the outcome?

- Recently Werner et al demonstarted that specific modalities of exercise might influence regulators of cellular senescence (i.e TL e TA). Please comment.

6. PLOS authors have the option to publish the peer review history of their article (what does this mean?). If published, this will include your full peer review and any attached files.

Reviewer #1: Yes: Jose-Angel Perez-Rivera MD,PhD

Reviewer #2: No

---

## [Author Response · Author response to Decision Letter 0]

12 Dec 2019

The manuscript has now been formatted to PLOS ONE’s style requirement.

Captions has been added and in-text citations updated accordingly. 

The research is supported by the National Institute for Health Research (NIHR) Newcastle Biomedical Research Centre based at Newcastle-upon-Tyne Hospitals NHS Foundation Trust and Newcastle University. VK has received research funding from the British Heart Foundation (CS/15/7/31679). The views expressed are those of the authors and not necessarily those of the NHS, the NIHR or the Department of Health.

Please remove any funding-related text from the manuscript and let us know how you would like to update your Funding Statement. Currently, your Funding Statement reads as follows: Nil

Funding related text have been removed from the manuscript.

4. We note that one or more of the authors are employed by a commercial company: Sanofi Genzyme.

The research is supported by the National Institute for Health Research (NIHR) Newcastle Biomedical Research Centre based at Newcastle-upon-Tyne Hospitals NHS Foundation Trust and Newcastle University. VK has received research funding from the British Heart Foundation (CS/15/7/31679). WQ is salaried by Sanofi Genzyme.

The funder provided support in the form of salaries for authors VK and WQ but did not have any additional role in the study design, data collection and analysis, decision to publish, or preparation of the manuscript. The specific roles of these authors are articulated in the ‘author contributions’ section.

Cover letter has been updated accordingly.

None of the authors have any conflicts of interests to declare. VK has received research funding from NIHR BRC Newcastle, AstraZeneca, and British Heart Foundation. WQ is salaried by Sanofi Genzyme. This does not alter our adherence to PLOS ONE policies on sharing data and materials.

Cover letter has been updated accordingly.

There are ethical restrictions on sharing a de-identified data set, as data contain potentially sensitive information as imposed by study sponsor and consent for data sharing not obtained from patients. Data are available upon request to the study Sponsor tnu-tr.sponsormanagement@nhs.net.

6. We note you have included a table to which you do not refer in the text of your manuscript. Please ensure that you refer to Tables 1, 2 and 3 in your text; if accepted, production will need this reference to link the reader to the Table.

This has now been revised in the manuscript.

Additional Editor Comments:

The Authors performed unadjusted cox regression models due to the lack of statistically significant difference between tertiles of TL. However, the sample size is small and this could led not to identify any statistically significant difference in baseline characteristics. Thus, I suggest a stepwise approach to be able to provide adjusted models, although incl. a limited number of potential confounders.

Multivariate cox regression models have been carried out and included in the manuscript. Nor telomere length or telomerase activity was predictive of primary outcomes after adjusting for age, gender and frailty. 

Reviewer's comments:

Reviewer #1: 1. Introduction: Previous data is well summarized and the main aim of the study is clearly specified.

 2. Methods:

 -Patients were selected from ICON1 study. Authors affirmed that patients aged > 65 years old were included. Maybe this cut-off point does not reflect the real “old patients”, in addition mean age of the patients included in this study is around 80 years old. Did you selected very old patients from the whole sample of ICON1 study or you included any patients older than 65 years old?

We agree that older patients would ideally have a greater age cut off. However, we believe that the mean age of participants in the study of 80 years does reflect older patients within this population. We did not specially select very old patients in this study, but included all patients older than 65 with available data for analysis from ICON-1 study. 

 -You divided the sample in tertiles. This is completely correct but it might be more precise to do it according to the best cut off point calculated by ROC curve. In this way you can provide data of sensitivity and specificity.

We have performed the analysis by dividing the predictors based upon area under curve (ROC). For TL, area under curve was 0.57 (p = 0.17). The best cutoff was 0.61 with sensitivity of 87% and specificity of 80%. With this cut-off, 111 participants had STL (82.2%) and 24 had LTL (17.8%). The area under curve was also measured for TA at 0.54 (p =0.57), with cut-off of 1.88 (sensitivity of 80% and specificity of 64%). 47 (70.1%) participants classified as low TA and 20 (29.9) as high TA. Telomere and telomerase activity were not predictive of primary outcomes. The results can be found in Table S5.

 -You designed a composite endpoint with thrombotic and bleeding events. In previous studies, telomere length seems to be connected to atherosclerosis and the evidence about its relationship with bleeding is weaker. It could be interesting to calculate a secondary combined endpoint formed only by thrombotic events.

We have performed the analysis using thrombotic events only as a secondary combined outcomes. The results suggest that Telomere and telomerase activity were not associated with the secondary outcomes. Details of this result can be seen in table S6.

 3. Results

 -The incidence of adverse events is too low in your patients. Previous studies of very old patients with ACS showed 6 months mortality around 12% (Alegre et al. J Am Med Dir Assoc. 2017 Nov 17).

We agree that the incidence of adverse events in our study is relatively low, which is highlighted in the study limitations. 

 -Please, comment the percentage of AMI and unstable angina in your sample.

Of the study participants, 109 (82%) with admitted with AMI, and 26 (18%) presented with unstable angina. Index presentation has been added to table 1.

 -Could you provide the incidence of mortality caused by cardiovascular causes in your sample?

Unfortunately, we did not have the detail of the specific cause of mortality in this study population. 

-Do you have data about the characteristics of the coronary event of your patients (number of affected arteries, Killip class, number of stents used…)? Is the peak value the troponin measurement provided en table 1?

Procedural data has now been added and can be found in table 2. Peak troponin measurements can be found in table 1. Telomere length was not associated with the characteristics of the coronary event in the study population. 

 4. Discussion

 --In the reference 25 of your manuscript (Epel et al.), telomere attrition but not initial telomere length was associated to cardiovascular mortality. In your study you did not find any relationship between telomerase activity and mortality. Could you explain in detail how your results agree or disagree with this paper?

Epel and colleagues found that the rate of telomere shortening was predictive of mortality from cardiovascular disease in elderly men, suggesting that low telomerase activity contribute to cardiovascular risk. However, our study found that telomerase activity was not associated with adverse outcomes following invasive management of NSTEACS. This may partly be explained by the difference of study populations. Epel et al. recruited healthy older volunteers prior to cardiovascular event, compared to the patients presenting with NSTEACS in the current study. Hence, this study focused on telomere length in those whom already developed coronary artery disease. The results together would suggest that dependent upon the cumulative oxidative stress and inflammatory burden, telomere attrition may play a different role in the process of atherosclerosis in older patients. 

-Other papers have studied the prognosis effect of telomere length in older patients with ACS (Am J Cardiol 2014;113:418e421). In this paper telomere length showed a prognostic effect in middle aged men with ACS but not in older patients. The group of elderly patients in this study was small (n: 52), presented STEMI and NSTEMI and revascularization was performed only in the 53% of patients so the sample is quite different to yours. Nevertheless, results are similar so it might be interesting to discuss this paper in your manuscript.

Perez-Rivera et al. found worse prognosis with short telomere length in middle aged men with ACS but not in older patients, which is consistent with the result of the current study. This would add further evidence that telomere attrition may play a different role in the process of atherosclerosis in older patients. This have been added to the discussion section of the manuscript. 

Reviewer #2: The Authors presented a substudy of the ICON-1 Biomarker study, aiming to determine the possible Association between telomere length and telomerase activity with the outcome of old patients presenting with NSTEACS.

 The manuscript is well written and addresses a clinically relevant issue with potential translational relevance.

 However, there are major issues to be resolved before considering the paper subitable for pubblication.

 - was this substudy pre-specified in the trial design? If not, this could be a relevant methodological bias.

This substudy was planned in the ICON-1 study, and has now been included in the methods section.

 - are the patients recruited consecutive?

In the ICON1 study, consecutive sampling was used. Participants were recruited from patients referred to two tertiary cardiac care hospitals from the neighbouring district general hospitals for invasive treatment of NSTEACS.

 - the reason why almost half of the patients recruited in the general study have been esclude in this substudy is clear and not reported. Please add. It is also relevant to explain Why Only in 67 patients telomerase activity data were available. These may be considered profound bias in the analysis of the data presented.

Only 135 participants had appropriate samples for testing telomere length and due to lack of funding, only 67 of all available samples could be analysed for telomerase activity. 

- The Authors stated that these patients underwent an invasive management of the NSTEACS. How many of these underwent PCI? Procedural data are not reported. Therapy on admission and at discharge was not reported. All these informations might have a profound impact on the data and outcome analysis.

A total of 111 (82.2%) patients undergone PCI, with 85 (63%) patients having single vessel disease, and 27 (20%) with multi vessels disease. Further details of the procedure and medical therapy on discharge has now been added and can be found on table 2. No significant association was observed between telomere length and procedure or medical therapy. 

 - the division in to tertiles seems to be arbitrary.

Telomere lengths was divided into tertile as it allows for the analysis between shorter and longer telomere length, which is consistent with previous literature.

 - When considering frail vs pre-frail patients, are there any difference in the distribuition of TL and TA and their association with the outcome?

As shown in table 1, there were no significant differences in telomere length between frail vs pre-frail patients. This is also observed in telomerase activity. Frailty was associated with outcome, as shown in the ICON1 study.

 - Recently Werner et al demonstarted that specific modalities of exercise might influence regulators of cellular senescence (i.e TL e TA). Please comment.

Werner and colleagues found that specific exercise modalities were associated with increased telomere length and telomerase activity. This raises the interesting possibility that the benefits of exercise in reducing and preventing frailty in the older population may act through regulating cellular senescence. The current study found no association between telomere length and frailty, which is consistent with the results demonstrated by Brault et al. in older adults with cardiac disease. These results suggest that frailty may be the consequence of repeated physiological stress induced premature cellular senescence in addition to progressive telomere shortening. It would be interesting to investigate the effect of exercise on telomere length and frailty in older adults with established cardiac disease. This discussion has been added into the manuscript under the discussion section.

---

## [Decision Letter · Decision Letter 1]

26 Dec 2019

The association of telomere length and telomerase activity with adverse outcomes in older patients with non-ST-elevation acute coronary syndrome

PONE-D-19-23361R1

Dear Dr. Kunadian,

We are pleased to inform you that your manuscript has been judged scientifically suitable for publication and will be formally accepted for publication once it complies with all outstanding technical requirements.

With kind regards,

Gianluigi Savarese

Academic Editor

PLOS ONE

Additional Editor Comments (optional):

Reviewers' comments:

Reviewer's Responses to Questions

**Comments to the Author**

1. If the authors have adequately addressed your comments raised in a previous round of review and you feel that this manuscript is now acceptable for publication, you may indicate that here to bypass the “Comments to the Author” section, enter your conflict of interest statement in the “Confidential to Editor” section, and submit your "Accept" recommendation.

Reviewer #1: All comments have been addressed

Reviewer #2: All comments have been addressed

2. Is the manuscript technically sound, and do the data support the conclusions?

Reviewer #1: Yes

Reviewer #2: Yes

3. Has the statistical analysis been performed appropriately and rigorously? 

Reviewer #1: Yes

Reviewer #2: Yes

4. Have the authors made all data underlying the findings in their manuscript fully available?

Reviewer #1: Yes

Reviewer #2: Yes

5. Is the manuscript presented in an intelligible fashion and written in standard English?

Reviewer #1: Yes

Reviewer #2: Yes

6. Review Comments to the Author

Reviewer #1: Thank you very much for your response. You have adequately addressed my suggestions. Congratulations for your research.

Reviewer #2: (No Response)

7. PLOS authors have the option to publish the peer review history of their article (what does this mean?). If published, this will include your full peer review and any attached files.

Reviewer #1: Yes: Jose-Angel Perez-Rivera

Reviewer #2: No

---

## [Editor Report · Acceptance letter]

31 Dec 2019

PONE-D-19-23361R1 

The association of telomere length and telomerase activity with adverse outcomes in older patients with non-ST-elevation acute coronary syndrome 

Dear Dr. Kunadian:

I am pleased to inform you that your manuscript has been deemed suitable for publication in PLOS ONE. Congratulations! Your manuscript is now with our production department. 

With kind regards,

on behalf of

Dr. Gianluigi Savarese 

Academic Editor

PLOS ONE